# Qualitative systematic review of general practitioners' (GPs') views and experiences of providing postnatal care

Clare Macdonald [1], Becky MacGregor [2], Sarah Hillman [2], Christine MacArthur [1], Debra Bick [3], Beck Taylor [1]

¹Institute of Applied Health Research, University of Birmingham, Birmingham, UK
²Unit of Academic Primary Care, University of Warwick, Coventry, UK
³Warwick Clinical Trials Unit, University of Warwick, Coventry, UK

**Correspondence to**
Dr Clare Macdonald;
ccm011@student.bham.ac.uk

## ABSTRACT

**Objectives** Develop an understanding of the views and experiences of general practitioners (GPs) about their role in postnatal care, including barriers and facilitators to good care, and timing and content of planned postnatal checks.

**Design** Qualitative systematic review.

**Data sources** Electronic database searches of MEDLINE, EMBASE, CINAHL, PubMed, Web of Science, PsychINFO from January 1990 to September 2021. Grey literature and guideline references from National Institute of Health and Care Excellence, WHO, International Federation of Gynecology and Obstetrics, Royal College of General Practitioners, Royal College of Obstetrics and Gynaecology.

**Inclusion criteria** Papers reporting qualitative data on views and experiences of GPs about postnatal care, including discrete clinical conditions in the postnatal period. Papers were screened independently by two reviewers and disputes resolved by a third reviewer.

**Quality appraisal** The Critical Appraisal Skills Programme checklist was used to appraise studies.

**Data extraction and synthesis** Thematic synthesis involving line-by-line coding, generation of descriptive then analytical themes was conducted by the review team. The Capability, Opportunity, Motivation-Behaviour (COM-B) model was used to develop analytical themes.

**Results** 20 reports from 18 studies met inclusion criteria. Studies were published from 2008 to 2021, reporting on 469 GPs. 13 were from UK or Australia. Some also reported views of non-GP participants. The clinical focus of studies varied, for example: perinatal mental health, postnatal contraception. Five themes were generated, four mapped to COM-B: psychological capability, physical opportunity, social opportunity and motivation. One theme was separate from the COM-B model: content and timing of postnatal checks. Strong influences were in physical and social opportunity, with time and organisation of services being heavily represented. These factors sometimes influenced findings in the motivation theme.

**Conclusions** GPs perceived their role in postnatal care as a positive opportunity for relationship building and health promotion. Addressing organisational barriers could impact positively on GPs' motivation to provide the best care.

**PROSPERO registration number** 268982.

## BACKGROUND

UK primary healthcare is generally organised and provided by practices who operate as independent contractors to the National Health Service (NHS), delivering 'General Medical Services'[1] to their registered patients. Since 2020, as part of this contract, providers in England are required to offer all women a postnatal check appointment 6–8 weeks after birth.[2] Community postnatal care, at home following birth, is multidisciplinary; as a minimum, midwives, health visitors and general practitioners (GPs) are typically involved. GPs see women for unplanned care in the postnatal period and for the routine 6 to 8-week check. Historically, in the UK, postnatal care provided after discharge from hospital has been an area of low priority in maternity services. Successive reports have found that women's experiences are often poor,[3 4] measurable outcomes such as breast feeding rates are low[5] and even national guidance contextualises recommendations by describing postnatal care as a 'Cinderella subject'.[6] This is in spite of the postnatal period being a high-risk time for maternal mortality and the unplanned maternal readmission rate being around 3.3%.[7]

A multidisciplinary, community-based postnatal care model is not unique to the UK. A systematic review of guidelines of community postnatal care found comparable recommendations from Australia and the USA.[8] More recently, Canadian public health guidance included similar principles and themes and,

similar to the UK, stated that multiple healthcare professionals will be involved with postpartum care.[9] In its position statement from 2021, the Royal Australian College of General Practitioners stated that there were gaps in evidence for the role of GPs in maternity care with more research needed.[10]

## STRENGTHS AND LIMITATIONS OF THIS STUDY

⇒ This review synthesised the views and experiences of general practitioners (GPs) about postnatal care, including its entire remit.
⇒ The search strategy was comprehensive, including grey literature and citation searches.
⇒ Only reports in English were included so some potentially relevant studies could have been overlooked.
⇒ Some included studies reported on non-GP participants in addition to GPs and it was not always possible to be certain which data was attributable to GPs.

UK GPs have become less involved in routine maternity care over time, markedly declining since the 1990s when intrapartum involvement dramatically reduced.[11] GPs are now often absent from antenatal care with women self-referring to NHS midwifery services, and midwives providing antenatal care even for women who subsequently require an obstetrician.[12] GPs have generally continued seeing women routinely around 6–8 weeks postnatally even when from 2004 to 2020 they had no contractual obligation to do so.[13] The quality was, however, variable; a 2017 report found women often felt rushed and 42% of those with emotional or mental health problems did not have this identified by their GP or other health professionals.[14] This report called for designated funding for the GP postnatal check and the Royal College of General Practitioners (RCGP) echoed this.[15] In 2020, the review of the NHS England GP contract included specific funding for the GP 6–8 weeks.[2] Soon after this, the National Institute for Health and Care Excellence (NICE) published its updated Postnatal Care Guideline[6] specifying that the check should be done by a GP.

The volume and variety of topics that should be covered in the maternal postnatal check, according to NICE is vast, even with no patient-related complexity. Factors such as mode of birth, intrapartum complications and comorbidities can affect the content of the consultation. Crucial to informing GPs' knowledge and understanding of the complexity of women's postnatal care requirements is information about their antenatal and intrapartum health and care provided by the hospital. NICE recommends that communication between secondary (ie, hospital) and primary care is prompt and effective, including a plan for ongoing care and conditions that require long-term management. Current arrangements for the maternal postnatal check do not distinguish formally between women who require more complex, or additional follow-up due to health and/or potential social care complications from those who do not.

To the authors' knowledge, no previous review has evaluated the views and experiences of GPs on postnatal care, incorporating the various clinical topics that form the 6–8 week maternal check as defined by NICE. There are examples of qualitative systematic reviews on discrete postnatal care topics, for example, gestational diabetes[16] and promotion of a healthy lifestyle in postpartum women,[17] but these do not provide an in-depth qualitative synthesis of the views and experiences of GPs encompassing the whole remit of postnatal care. The Cochrane Collaboration has planned a qualitative systematic review of factors influencing the provision of postnatal care by health workers,[18] but this will also include secondary and tertiary care settings with no specific focus on GPs' perspectives.

Women's experiences are better documented than GPs' experiences and suggest inconsistencies in provision of care.[19] Given the broad scope and variability in quality of the GP 6–8 week maternal postnatal check, in order to understand the dynamic of these consultations and potential practitioner-based facilitators or barriers, it is necessary to understand the contextual and behavioural basis that potentially impacts the provision of postnatal care by GPs.

## AIMS

This qualitative systematic review aimed to answer four research questions:

1. What do GPs say is, or should be, the content of the routine 6–8 week check (including non-clinical topics such as social complexity)?
2. At what point in time do GPs say that they routinely review women after birth, and what timing do they think is optimal?
3. What do GPs see as the facilitators and barriers to providing high-quality postnatal care in primary care settings?
4. What do GPs perceive to be their role in postnatal care, and what role do they say other members of the primary care team have or should have?

## METHODS

### Inclusion and exclusion criteria

The PerSPecTIF framework[20] was used to develop and define the detail of the research questions, and therefore eligibility criteria for inclusion of studies (table 1). Given variability in provision of postnatal care in primary care, particularly the postnatal check, this framework was chosen with the aim of achieving a better understanding of context and acceptability from the perspectives of individuals about care provision. There was no intervention evaluated, or comparison made.

### Information sources

Database searches of MEDLINE, EMBASE, CINAHL, PubMed, Web of Science and PsychINFO were carried out in September 2021. Reference lists of included studies were reviewed for other potential studies meeting inclusion criteria, as were references from relevant grey literature including guidance from NICE, WHO, International Federation of Gynecology and Obstetrics, RCGP and Royal College of Obstetrics and Gynaecology. Citation searching of included publications was completed. Grey literature and citation searches were conducted in March 2022.

### Search strategy

Searches were limited to publications where full text was available in English. Studies from before 1990 were excluded since UK GP involvement in maternity services became more confined to postnatal care from that time. Studies with populations from low-income

**Table 1** PerSPecTIF table of inclusion and exclusion criteria

| PerSPecTIF | Inclusion | Comments |
|---|---|---|
| **Per**spective | General practitioners<br>General practitioner specialty trainees (GPSTs) (sometimes termed registrars)<br>Primary care physicians | Exclude hospital physicians/obstetricians/paediatricians<br>Exclude other healthcare workers, for example, midwives, health visitors, pharmacists |
| **S**etting | Primary care | Exclude secondary care settings |
| **P**henomenon of interest | Postnatal care of women and routine postnatal checks for women | Including the following defined clinical aspects of postnatal care<br>► Postnatal contraception<br>► Postnatal mental illness<br>► Gestational diabetes<br>► Hypertensive disorders of pregnancy<br>► Urinary incontinence<br>► Pelvic organ prolapse<br>Including, but not limited to other relevant topics such as postnatal infections (mastitis, perineal wound infection, caesarean section wound infection), breast feeding related problems<br>Including health promotion, social issues or concerns, social complexity, social prescribing |
| **E**nvironment | Community settings providing planned (routine review or follow-up) and unplanned postnatal care from any GPs | |
| **(C)**omparison | No comparison | |
| **T**iming | The postnatal period, encompassing the first 2 years after giving birth (because perinatal mental health services accept referrals up until 2 years after birth) | |
| **F**indings | In relation to the views and experiences of GPs, with relevance to GPs, policy makers, medical educators | |

GP, general practitioner.

and middle-income countries[21] were excluded to maintain relevance to the UK setting. The search strategy was piloted to ensure it captured key studies known to reviewers (see online supplemental file 1).

## Screening and data extraction
After the removal of duplicates, titles and abstracts of all records were independently screened by two reviewers (CM and SH or BM). A third reviewer (BT) screened records where there was disagreement. A standardised data extraction proforma was developed to extract consistent, relevant information from studies including year of publication, setting and analysis methodology. Data were extracted in line with the approach described by Thomas and Harden[22] whereby anything reported as 'findings', 'results' or equivalent were included in the analysis. Data from non-GP participants were not extracted. Data included themes, descriptions of themes and participant quotes. Full texts, including abstracts were checked for anything that could be considered a study findings. CM and SH or BM independently completed data extraction for a third of the included studies and there was a high degree of consistency, CM then completed data extraction for the remaining studies.

## Data synthesis
The thematic synthesis approach outlined by Thomas and Harden[22] was used as the basis for data synthesis. After multiple readings to gain familiarity, CM imported extracted data into NVivo12 software. Line by line coding was undertaken by CM, with BT independently coding a third of data and consistency of approach confirmed. CM and BT, in discussion with other review team members developed descriptive codes which was inductive, but with reference to the research questions. The data synthesis process was iterative and collaborative, taking place over several weeks with continual reference to the original studies and input from the review team to ensure that interpretation reflected the original studies. The review team remained open to the possibility of building the analytical themes around pre-existing frameworks or models, depending on the nature of the data and the descriptive themes identified.

## Patient and public involvement

The focus of this review was informed by feedback from the Applied Research Collaboration, West Midlands (ARC-WM) PPIE group that experiences of postnatal care, including the GP check were sometimes poor. This review was completed as part of a wider project around improving the GP postnatal check for which the ARC-WM PPIE group have regularly contributed.

## RESULTS

### Included studies

Two thousand six hundred and ninety-three unique references were identified in database searches and following screening, 23 full-text reports were assessed for eligibility by CM. The review team agreed the final set of included studies. Seventeen reports describing 15 studies met inclusion criteria from this process. One excluded report was an integrative review on physician perspectives of their role in perinatal mental health.[23] Studies included in this report were screened and two of these met eligibility criteria. Reasons for exclusion from the database searches were incorrect phenomenon of interest[24 25] or incorrect perspective.[26–30] Three studies from references or citations of eligible reports met the inclusion criteria and were also included. In total, 20 reports were eligible, reporting on 18 studies (figure 1).[31–50]

Studies were published between 2008 and 2021 representing views and experiences of 469 GPs. Nine were UK studies, four Australian and the remainder from other countries, all with some degree of similarity to the UK postnatal care model. Twelve studies included other clinicians and GPs (eg, health visitors, obstetricians), six included GPs only. As anticipated, studies focused on a range of postnatal care topics, or postnatal care overall. Most frequent were perinatal mental health (n=7) and follow-up of gestational diabetes (n=4). The remainder investigated postnatal care generally, follow-up of hypertension or pre-eclampsia, behaviour change opportunities, contraception and prescribing for breastfeeding women. Two studies reported qualitative analysis of survey results, others used various forms of interview for data collection; the inclusion of qualitative analysis of survey data explains the large number of participants overall (see online supplemental file 2).

### Quality appraisal

Quality appraisal was undertaken using the Critical Skills Appraisal Programme (CASP) Qualitative Checklist.[51] Studies were not excluded on methodological merit, but CASP was used to formally understand their integrity and reliability which aided interpretation of findings during the synthesis. CM completed the CASP assessment, with oversight from the review team. Most studies fulfilled all or nearly all the checklist criteria, the area with least compliance being consideration of the relationship between researcher and participants which was only fulfilled in five studies (see online supplemental file 3).

## Themes

Sixteen descriptive themes were identified (see online supplemental file 4). The final stage of thematic synthesis involved identifying analytical themes; moving beyond a descriptive summary of aggregated findings. One descriptive theme related specifically to two of the research questions—content and timing of the postnatal check. This did not generate new, analytical findings and is summarised descriptively. The remainder of the descriptive themes related to influences, barriers and facilitators to GPs in providing postnatal care. When reviewing and discussing these descriptive themes, to move to analytical themes, the research team reflected that these factors could in fact be mapped to the COM-B (Capability, Opportunity, Motivation-Behaviour) model[52]; this became the basis on which the final analytical stage was completed. The descriptive themes were reviewed with reference to COM-B and mapped to relevant aspect of the model. The final themes were therefore: content and timing of postnatal checks (not mapped to COM-B), psychological capability, physical opportunity, social opportunity and motivation (no descriptive themes were mapped to the COM-B domain of 'physical capability') (figure 2) (see online supplemental file 5).

### Content and timing of postnatal checks

Most studies mentioned the content of a routine postnatal check, or the typical remit of GPs' interactions with women.[31 32 35–37 39–42 44–46 48–50] A broad range of specific clinical topics were cited as falling under GP responsibility, including asking about complications, such as hypertension or diabetes, more universal issues such as mental health, bleeding, pain, wound healing, fever, breast feeding, sleep, resumption of sexual intercourse, contraception, future pregnancy plans, advice about exercise, diet and weight loss, and reminders about cervical screening. There was a strong sense that GPs perceived the routine postnatal check as holistic, with references to building a relationship with the woman's wider family[32 44 49] and preventative medicine[37 45 46 49]:

*The GPs interviewed did not see their role limited to routine checks and managing illness. Instead they thought their responsibility extended to assessing how the family was functioning and providing anticipatory guidance and education when appropriate.[32]*

GPs' views on when a routine postnatal check should happen, and whether there should be more than one planned contact, were not commonly discussed. Two studies made specific reference to this[37 44] without consensus regarding the optimal time; the earliest time discussed was a review within 2 weeks after birth[32] and no studies described a recommendation later than 8 weeks. There was also a suggestion that the timing was woman-dependent and that 6 weeks may be right for some, but not all women. Similarly, the duration of the appointment was variable depending on the needs and complexity of the woman:

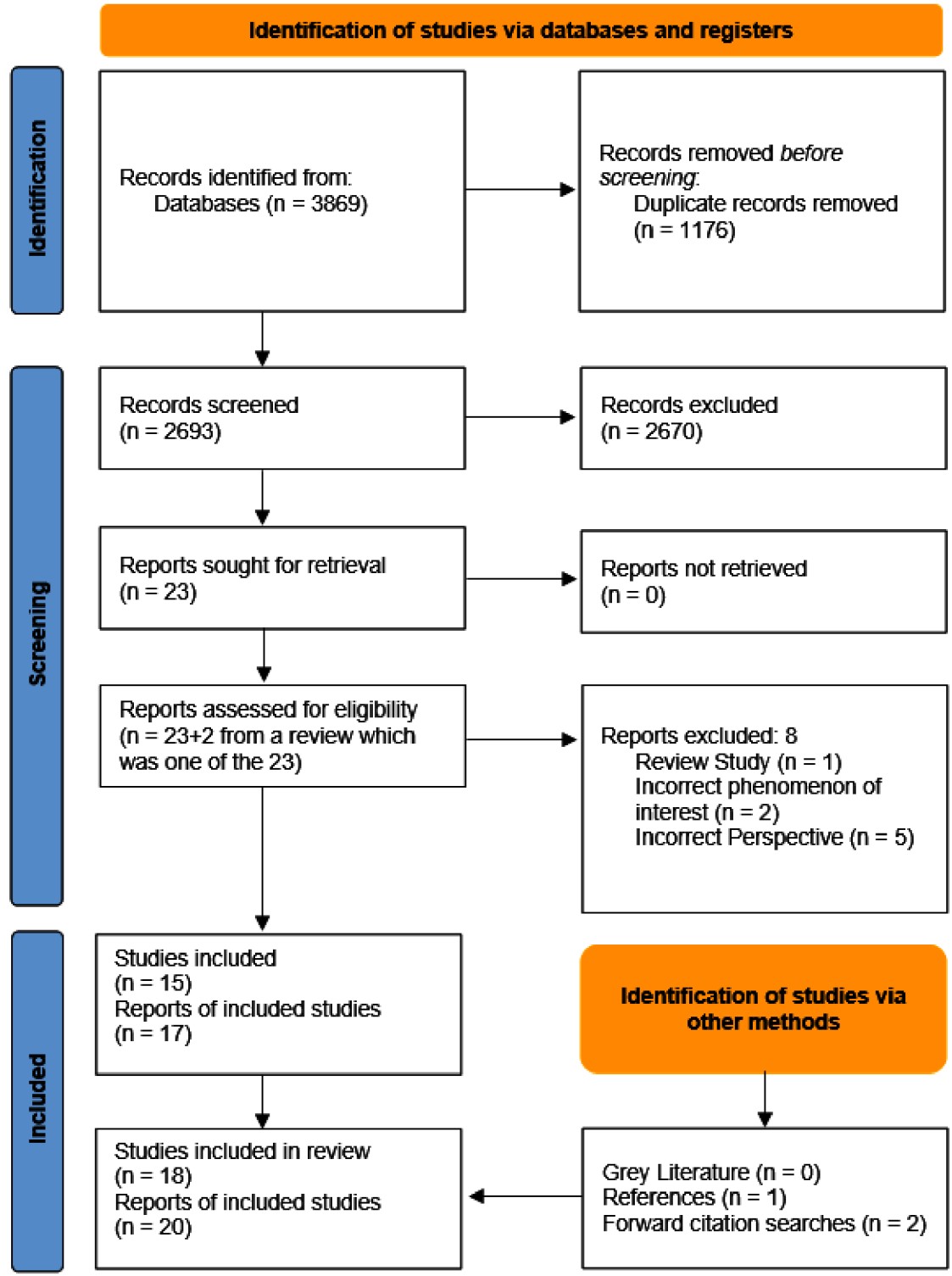

**Figure 1** Study selection process.

*six of the GPs felt that it [6 weeks] was potentially too late for young mothers: "I guess you could argue it's potentially a bit late for some women. I can certainly think of one or two quite young women who have got pregnant very quickly after having the first baby, and you wonder whether you might in some cases miss it". [GP11]*[42]

## Influences on GPs' approach to postnatal care

### Psychological capability

Psychological capability encompasses an individual's mental functioning, including understanding and memory.[52] A broad range of knowledge was found to be required to provide postnatal care which could be

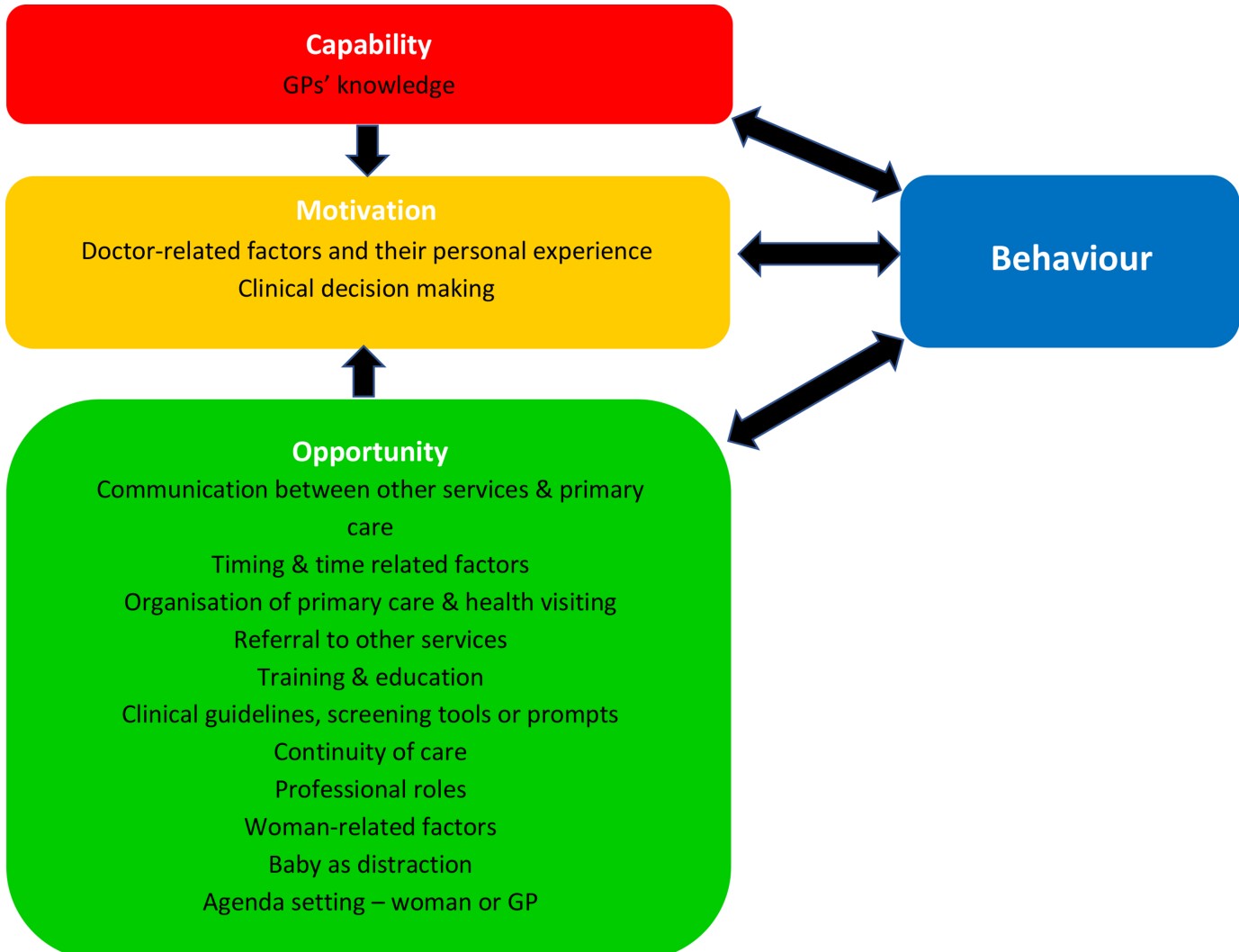

**Figure 2** Themes mapped to Capability, Opportunity, Motivation-Behaviour. GP, general practitioner.

explained by the variety of clinical topics included, but one study focusing on postpartum care in general found good self-reported knowledge of a basic postpartum review:

*Most PCPs [primary care physicians] claimed to know the basic clinical examination of a postpartum woman. They were aware on the assessment of lochia cessation, wound recovery, pain control, screening for post-natal depression, breastfeeding, and preventive care, such as cervical cancer screening and contraception advice.*[46]

This was accompanied by acknowledgement that having a good basic knowledge as a generalist meant it was sometimes difficult to be expert in discrete areas:

*Participants acknowledged that the difficulty for GPs is that they offer a generalist service making it difficult for them to have expertise in all areas including PMH [perinatal mental health]: 'You are kind of jack-of-all-trades, you are master of none*[44]

There was overlap between psychological capability and physical opportunity in that GPs often referred to their

use or awareness of resources such as guidelines when addressing the issue of their knowledge. Recall and interpretation of guidelines was sometimes poor.

*… you're not really sure what the responsibility is or what the frequency with which you should be checking it [blood pressure] is, and I don't think that the NICE guidance is so clear on that. (Participant 11—General Practitioner)*[31]

Some knowledge gaps were identified in studies which explored specific topics, for example, awareness of the condition itself,[47] how a condition impacts the woman's mental health[48] and future risks.[50] These findings across different conditions demonstrated the vast spectrum of pathology that GPs may come across postnatally and highlights that knowledge of conditions was not universal.

*Some HCPs [healthcare professionals] were not aware of PNA [perinatal anxiety] as a specific diagnosis and were uncertain that PNA existed as a distinct clinical entity: 'I mean it's a concept that I'm really not that much aware of either professionally or from reading'.*[47]

The knowledge required to fulfil the broad remit of postnatal care was extensive and GPs expressed they had a good basic grasp of core topics but sometimes lacked knowledge of specific clinical areas, including difficulty understanding, accessing or interpreting guidelines.

## Physical opportunity

This theme describes environmental and physical resources and context with reference to the action or behaviour in question[52] and was represented by all studies.[31–50] Issues such as length of time available, guidelines and screening tools, and information supplied were frequently discussed. Some elements that were representative of both physical and social opportunity are summarised here.

GPs relied on information received from other services (usually hospital) about antenatal and intrapartum journeys. Information was sometimes inadequate, whether about a specific complication or care in general. There was a desire to have improved information, with frustration expressed by words such as 'disappointing', and descriptions of difficulty accessing relevant information quickly.

*GPs also identified the need for clarity and explicit recommendations for postnatal follow-up, indicating that the process and communication could be improved: "I would be keen to do a little bit more if it was simple and clear". [GP9][41]*

Guidelines, toolkits and screening tools were cited as resources known to or used by GPs as part of their clinical practice or continuing professional development. There was criticism of the clarity, accessibility and usability of guidelines across the different clinical topics which were out of the control of individual GPs.

*Non-availability of easily accessible, evidence based, up to date information on medicines in breastfeeding was mentioned. GPs often mentioned that their sources of information were conflicting and often "over cautious".[38]*

Exposure to perinatal learning experiences and training were described as variable, in terms of how much training GPs had overall, and that some clinical topics such as breastfeeding were not as well covered as others.

*Participants recalled limited information related to PMH [perinatal mental health] as the focus of their specialist training was on obstetric and medical topics and in terms of practical experience: 'It can be luck of the draw what you do get exposed to'. (P1)[44]*

There was no clear or consistent view on how long was allocated or needed for a postnatal check. The complexity of the woman impacted on how long the appointment would take, and therefore whether all aspects of the check were afforded adequate time. Moreover, there was sometimes a tendency to avoid certain clinical topics or tasks due to a lack of time, or fear of something time-consuming emerging. GPs would sometimes arrange a follow-up appointment if not everything could initially be covered.

*Most felt that a more realistic time per consult should last from 15 to 20 min, compared to the current 5 to 10 min in real-time clinical practice. The limited time for consult often reduced the comprehensiveness and scope of postpartum issues covered within the postpartum visit. Consequently, the PCPs [primary care physicians] tended to avoid time-consuming tasks such as mental health assessment or discussing contraception options with the mothers.[46]*

In UK studies in particular, there was frustration with organisational changes meaning that health visitors and GPs were no longer colocated, impacting negatively on communication. There were positive descriptions of examples of good care when services were organised as well run multi-disciplinary teams. GPs sometimes felt disconnected by their reduced role in antenatal care which impacted on their ability to provide continuity. Organisational changes around GP and health visitor working were described as reducing opportunities for long-term relationships. However, GPs reflected positively on their professional autonomy to offer follow-up appointments when indicated. There was a recognition of the difficulties that women could face in accessing primary care services.

*practitioners reported that ongoing NHS reform had resulted in health visitors being moved out of general practice and into centralized services (often in non-NHS settings), a development universally regarded as deleterious to multiagency team work and delivery of effective perinatal mental healthcare[36]*

Overall, many aspects of physical opportunity were consistently cited as important to GPs. The wide range of elements identified reflected the complexies of postnatal care itself and the intricacies of service organisation. That GPs sometimes had little or no influence over these factors impacted on their approach to patients with a sense that they wanted more time to be made available for the women with more complex issues, and better resources to complement their clinical experience.

## Social opportunity

Six descriptive themes were relevant to social opportunity, which encompasses aspects of social norms and culture involving interactions between different people.[52] There was sometimes uncertainty from GPs about who was responsible for setting the agenda in the postnatal check or ensuring appropriate follow-up. Some GPs described a flexible approach, covering both the woman's and the GP's agenda.

*You've got to be quite fluid …. What is it that they've come to see you about? I've got to meet that need primarily. I've got to discharge my need which is keeping them safe and running through all the possible scenarios of something going terribly*

*wrong … anything extra that I can plug in terms of education, that's a bonus. (RGP7)*[32]

Also influencing the nature of the consultation, was GPs citing the baby as a distraction, impacting on their care of the woman.

*The GPs reported that after delivery, the primary focus of care shifts from the woman with GDM to the baby.*[50]

GPs expressed a desire to understand the cultural and social background of women and how this might impact care they needed. Additionally, GPs sometimes contextualised potential barriers to compliance with care or recommendations within their perception of social norms about women finding it difficult to engage with their health and future risk.

*The attitudes of a mother to self-care and health were seen as a barrier. Denial of risk was perceived by some GPs to further hinder follow-up care.*[45]

Overlapping with organisational factors captured in the physical opportunity theme were GPs' perceptions of their own role, and roles of colleagues such as health visitors and midwives. GPs' minimal or absent involvement with women antenatally was cited as a reason for difficulties with postnatal care, or justification for some aspects of postnatal care being handled by someone else.

*A few GPs also referred in qualitative comments to their increasing distance from antenatal care due to organisational changes placing some midwives and health visitors outside practice surgeries: "I feel my role has been marginalised since joint working with health visitors has effectively stopped". (GP, survey respondent)*[39]

This theme captured tensions GPs experienced when trying to balance conflicting consultation agendas, distractions and evolving professional roles with their own beliefs about what women want or need postnatally.

## Motivation

The COM-B model distinguishes between reflective and automatic motivation,[52] but in the analysis, motivational features were difficult to categorise as solely one of these (COM-B acknowledges that, for example, automatic processes can arise from reflection[53]) and therefore Motivation was combined as a single theme. The descriptive themes that mapped to this were 'Doctor-related factors' and 'Clinical decision making'. This theme captured GPs' perceptions about how they reached decisions, and insights into past professional or personal experiences that impacted their actions. Studies with a focus on mental health captured examples of reticence to arrive at a diagnosis, a desire to normalise aspects of perinatal mental illness, including dismissing symptoms as being part of normal life.[33–35 39 43]

*[with reference to a case vignette of a traumatic birth] "Without taking anything away from the trauma and distress suffered, there is a chance that this also could be*

*something this woman could work through and doesn't necessarily see a doctor about and manages to recover from quite well without needing medical intervention". GP 6 (Male)*[43]

GPs sometimes resisted their instinct or known expectation to explore perinatal mental health issues for reasons such as desire to avoid awkwardness.

*Can I be honest with you sometimes I wonder if you really want to open this can of worms and it's so much easier just to jolly along and check the BP, check the urine, check this and that and have them out the door and see the next patient. (P5)*[44]

In some cases, social opportunity seemed to impact on motivation: the nature of relationships between healthcare professions or between GPs and women influenced the motivation of the GP to act in a certain way. For example, when continuity of care was not achieved, GPs were less motivated to be thorough:

*Some health professionals described consciously inhibiting disclosure in order not to be placed in this position citing lack of continuity of care as the reason: "Easier not to ask, if I'm not going to see her again". (L GP1)*[34]

Some factors appeared to increase GPs' confidence in making a diagnosis or enabling disclosure of symptoms such as involvement of others in decision making, use of clinical instinct, and patient insight into their symptoms.

*Your antenna would be raised by people coming clean with you that there is something going on … a history … the usual kind of joy isn't there, they're, quiet. (P5)*[44]

GPs' views about their roles most commonly fell within the social opportunity theme, but there were motivational influences whereby beliefs about their role could impact the nature of their interaction with the woman.

*Many considered their main purpose as treating physical health problems and not to promote health.*[49]

Overall, there was a reliance on experience and clinical acumen and a desire to share decision making with the woman and professional colleagues. This was combined with some resistance to completing a diagnostic process when pathology had been identified or even avoiding the diagnostic process altogether.

## DISCUSSION

This review suggested wide ranging influences in GPs' overall approach to postnatal care, the strongest being physical and social opportunity, and that these issues impact on motivational factors. With regard to the timing of routine checks, it was perhaps surprising that little data made reference to this, given its lack of evidence base and therefore potential for uncertainty. There was, however, some uncertainty about whether 6–8 weeks was the ideal time, for example, that it might be too late for contraceptive needs, but also an unquestioning acceptance that

6–8 weeks was the norm. This could be explained by 6–8 weeks being culturally embedded, being recommended in guidelines and coinciding with the planned infant physical examination and first immunisations. The absence of research to determine the optimal time for a routine postnatal check could explain why GPs lacked a clear sense of what might be ideal, and usual practice continuing without formal evaluation as to whether 6–8 weeks is appropriate to conclude a woman's maternity care. The content of postnatal care consultations, in terms of the clinical topics covered was described as wide reaching and was often influenced by GPs' professional experience or the woman's own agenda rather than formal clinical guidance. This introduced further variability because, for example, GPs had different levels of experience and were not always able to influence their own clinical jobs or rotations, and women's confidence, knowledge or ability to present their own agenda would vary significantly.

GPs perceived themselves as having a key role in the clinical care of women—and their families—in the postnatal period. In describing their perception of the roles of other healthcare professionals, they clearly expressed the benefits of multidisciplinary working, and the importance of the roles of others, particularly health visitor colleagues. It appeared that when there was physical proximity, or straightforward access to health visitors, GPs felt more able to provide better care. However, there was also reference to the difficulties arising from lost continuity with antenatal care falling outside of the GPs' remit, and complex women being managed in secondary care without clear communication. A key facilitator was the professional satisfaction and enjoyment GPs derived when seeing women postnatally, as part of their care for the whole family. However, they could be reluctant to provide a comprehensive postnatal check when constrained by time or perceived a risk of uncovering complexity by raising certain topics such as perinatal mental health. GPs viewed the task of providing care for women postnatally as more challenging when they lacked time, clear clinical guidance and access to suitable communication from secondary care.

Limitations of this review included the heterogeneity between studies which was anticipated but resulted in the synthesis of data from studies about different discrete clinical topics. This was considered throughout the analysis, and findings from studies concerning postnatal care as a broad topic contained similar findings, giving some assurance to this approach. The inclusion of non-GP participants in some studies meant that during data extraction, it was sometimes difficult to be certain which data was from GPs. Although there are examples globally of similar postnatal care models to the UK, there are important differences that could influence the analysis, for example, studies from Australia and Singapore reference different needs of private and public patients which is not so relevant to the UK. Finally, while the included studies were all appraised against CASP criteria, one was a report not formally peer-reviewed.[39]

Usage of the COM-B model as part of the analysis was made after the team reflected that this model was able to neatly frame what the data were showing. Application of the COM-B model to the synthesis allowed for translation of findings to the model's associated Behaviour Change Wheel (BCW)[52] and therefore the generation of potential intervention or policy areas that apply to each behaviour. The strongest influences of social and physical opportunity revealed intervention categories of education, persuasion, incentivisation and coercion. GP practices have already been financially incentivised to offer women a postnatal check since the 2020 change to the NHS England GP contract, but this has not been accompanied by complementary interventions such as education or persuasion. It may be possible in the future to determine whether this intervention has improved postnatal check provision, but it will not measure quality. By targeting interventions recommended in the BCW from the motivation and social opportunity themes (such as enablement, training, education), quality and consistency could be improved.

The postnatal period is one of marked health inequality.[7] This is a critical issue that requires further research to better understand causes, and intervention to overcome it. This review found minimal reference to differential risk between populations, except for one study which looked specifically at black and ethnic minority women,[36] and higher risk groups were not well defined or explained in included studies. The suggestion within the physical opportunity theme that different women needed different lengths of time for a postnatal review may be important in addressing this inequality; if women at higher risk of morbidity or mortality could be identified, they could be given a longer appointment, or additional follow-up. From what is already know about disparities in outcomes for women with different social and medical backgrounds, it may be reasonable to extrapolate that adjusting routine care to meet needs has the potential to improve outcomes and reduce inequality.

Future research should address the issue of timing of the routine maternal postnatal check, both when it should happen, and how long it should take. These research areas should be approached with specific reference to groups known to be most at risk. In the UK context, the introduction of a mandatory maternal postnatal check to the GP contract should be evaluated, from the perspectives of women, but also GPs who provide it, so that its provision and quality are optimised.

**Contributors** CM developed the protocol, carried out the searches, participated in the selection process and data extraction, undertook the coding process, led the analysis, drafted the manuscript and is the guarantor. BM participated in the selection process and data extraction, contributed to the analysis, and contributed to and approved the manuscript. SH reviewed the protocol, participated in the selection process and data extraction, contributed to the analysis, and contributed to and approved the manuscript. CM reviewed and contributed to the protocol, contributed to the analysis, and contributed to and approved the manuscript. DB reviewed and contributed to the protocol, contributed to the analysis, and

contributed to and approved the manuscript. BT reviewed and contributed to the protocol, participated in the selection process, undertook the coding process, contributed to the analysis, and contributed to and approved the manuscript.

**Funding** CM is funded by the National Institute for Health Research (NIHR) West Midlands Applied Research Collaboration (ARC). The views expressed are those of the author and not necessarily those of the NIHR or the UK Department of Health and Social Care.

**Competing interests** None declared.

**Patient and public involvement** Patients and/or the public were involved in the design, or conduct, or reporting, or dissemination plans of this research. Refer to the Methods section for further details.

**Patient consent for publication** Not applicable.

**Ethics approval** Not applicable.

**Provenance and peer review** Not commissioned; externally peer reviewed.

**Data availability statement** Data are available upon reasonable request.

**ORCID iDs**
Clare Macdonald http://orcid.org/0000-0002-9947-2686
Becky MacGregor http://orcid.org/0000-0002-9530-8387
Sarah Hillman http://orcid.org/0000-0002-3560-2074
Christine MacArthur http://orcid.org/0000-0003-0434-2158
Debra Bick http://orcid.org/0000-0002-8557-7276
Beck Taylor http://orcid.org/0000-0002-3559-7922

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
