## [Reviewer comments · BMJ Open]

ARTICLE DETAILS

TITLE (PROVISIONAL)	A qualitative systematic review of general practitioners' (GPs') views and experiences of providing postnatal care.
AUTHORS	Macdonald, Clare; MacGregor, Becky; Hillman, Sarah; MacArthur, Christine; Bick, Debra; Taylor, Beck

VERSION 1 – REVIEW

REVIEWER	Nyondo-Mipando, Alinane Linda Kamuzu University of Health Sciences
REVIEW RETURNED	08-Dec-2022

GENERAL COMMENTS	This is a good review especially now when WHO has just released the new guidelines on positive postnatal care. The section that the authors need to beef up is the discussion which only includes a single paragraph where the findings are discussed. The authors should review the objectives and present a discussion that reflects on them. Otherwise, the rest of the review is well written and is a good read.
--

REVIEWER	Vieira, Elisabeth Meloni University of Sao Paulo, Saúde, Ciclos de Vida e Sociedade
REVIEW RETURNED	17-Jan-2023

GENERAL COMMENTS	The subject under study is important and relevant for women's health, approaching the views and experiences of General Practitioners in providing postnatal care. It is a systematic review of qualitative studies. The paper is well written. The objectives of the study are clearly defined. The method used is well described. A wide range of electronic data was searched, as well as the guidelines of relevant internationally recognized health organizations such as the World Health Organization, the Royal College of GP, and the International Federation of Gynecology and Obstetrics among others. The PerSPecTIF framework was used to develop and define the research questions and eligibility for the study. The inclusion criteria are clearly stated. Two reviewers screened the papers and the disputes were solved by a third one. The CASP (Critical Appraisal Skills Programme) checklist was used to assess the studies and the COM-B model was used as a theoretical framework to analyze the results organized by thematic analyzes. The results are coherent and consistent with the adopted framework. They can subsidize relevant changes in the GP contract and postnatal guidelines in the UK. The PRISMA checklist for systematic review was used and several supplements reinforcing the theoretical and empirical
---

	bases for the study are attached to the paper. I recommend the publication
--	--

VERSION 1 – AUTHOR RESPONSE

Reviewer: 1

Dr. Alinane Linda Nyondo-Mipando, Kamuzu University of Health Sciences Comments to the Author: This is a good review especially now when WHO has just released the new guidelines on positive postnatal care.

The section that the authors need to beef up is the discussion which only includes a single paragraph where the findings are discussed. The authors should review the objectives and present a discussion that reflects on them.

Otherwise, the rest of the review is well written and is a good read.

Thank you for your positive comments. The Discussion section has now been updated and expanded and now includes reference to each of the research objectives.

Reviewer: 2

Dr. Elisabeth Meloni Vieira , University of Sao Paulo Comments to the Author:

The subject under study is important and relevant for women's health, approaching the views and experiences of General Practitioners in providing postnatal care. It is a systematic review of qualitative studies. The paper is well written. The objectives of the study are clearly defined. The method used is well described. A wide range of electronic data was searched, as well as the guidelines of relevant internationally recognized health organizations such as the World Health Organization, the Royal College of GP, and the International Federation of Gynecology and Obstetrics among others. The PerSPecTIF framework was used to develop and define the research questions and eligibility for the study. The inclusion criteria are clearly stated. Two reviewers screened the papers and the disputes were solved by a third one. The CASP (Critical Appraisal Skills Programme) checklist was used to assess the studies and the COM-B model was used as a theoretical framework to analyze the results organized by thematic analyzes. The results are coherent and consistent with the adopted framework. They can subsidize relevant changes in the GP contract and postnatal guidelines in the UK. The PRISMA checklist for systematic review was used and several supplements reinforcing the theoretical and empirical bases for the study are attached to the paper. I recommend the publication

Thank you for your positive comments.